# Effects of Sting Plant Extracts as Penetration Enhancers on Transdermal Delivery of Hypoglycemic Compounds

**DOI:** 10.3390/medicina55050121

**Published:** 2019-05-07

**Authors:** Yuh-Ming Fuh, Dinh-Chuong Pham, Ching-Feng Weng

**Affiliations:** 1Department of Life Science and Institute of Biotechnology, National Dong Hwa University, Shoufeng, Hualien 97401, Taiwan; ming56564542@yahoo.com.tw; 2Faculty of Applied Sciences, Ton Duc Thang University, Ho Chi Minh City 700000, Vietnam; phamdinhchuong@tdtu.edu.vn

**Keywords:** transdermal, permeation enhancers, skin permeation, oral glucose tolerance test, diabetes mellitus

## Abstract

*Background and objectives*: The percutaneous route is an interesting and inventive investigation field of drug delivery. However, it is challenging for drug molecules to pass through the skins surface, which is a characterized by its permeability barrier. The purpose of this study is to look at the effect of some penetration enhancers on in vivo permeation of insulin and insulin sensitizers (curcumin and rutin) through diabetes-induced mouse skin. *Materials and Methods*: Sting crude extracts of *Dendrocnide meyeniana*, *Urtica thunbergiana* Sieb. and Zucc, and *Alocasia odora* (Lodd.) Spach were used as the penetration enhancers. Mouse skin irritation was tested by smearing the enhancers for the measurements at different time points and the cell viability of the HaCaT human skin keratinocytes, which was determined by Trypan blue exclusion and MTT assays to evaluate human biosafety for these extracts after the mouse skin permeation experiments. *Results*: All enhancers induced a slight erythema without edema on the mouse skin that completely recovered after 6 h from the enhancer smears as compared with normal mouse skin. Furthermore, no damaged cells were found in the HaCaT keratinocytes under sting crude extract treatments. The blood sugar level in the diabetic mice treated with the insulin or insulin sensitizers, decreased significantly (*p* < 0.05) in the presence of enhancers. The area under the curve (AUC) values of transdermal drug delivery (TDD) ranged from 42,000 ± 5000 mg/dL x min without enhancers, to 30,000 ± 2000 mg/dL x min in the presence of enhancers. *Conclusions*: This study exhibited that natural plant extracts could be preferred over the chemically synthesized molecules and are safe and potent penetration enhancers for stimulating the transdermal absorption of drugs.

## 1. Introduction

Transdermal drug delivery (TDD) has been recognized as a promising and painless approach with advantages such as favorable reception by patients (noninvasiveness), enhancing bioavailability, minimal toxic side effects, and ease of access [1,2]. However, skin is one of the primary challenges in the delivery of a majority of drugs via the transdermal route. The immense hindrance for TDD is found in the stratum corneum (SC) of the skin, which is mainly composed of insoluble keratinocytes (70%) together with lipids (20%) that reduce water evaporation from the dermis and blocks the permeation of chemical and biological drugs [3]. Many approaches have been shown to reshape SC barrier properties and to promote the absorption of molecules through the skin for gaining adequate drug levels at the fitting site. Among them, transdermal penetration enhancers (PE), e.g., terpenes [4] and surfactants [5] emerge as key elements to enhance the skin permeability by disrupting the complex structure of the SC lipid or combining with intercellular proteins. Although more than 300 molecules have been shown to facilitate the passage of molecules across the SC, few of them have been commercialized as transdermal drugs, due to compositional conflicts or irritation concerns [6]. For example, dimethyl sulphoxides (DMSO) is one of the most commonly studied PEs that are found to cause erythema and wheal of SC [7], or oleic acid in propylene glycol is found to have caused severe skin irritation in human volunteers [8]. Hence, searching the safe and highly efficacious enhancers for TDD, particularly from natural sources remains an important issue for pharmaceutics.

Diabetes mellitus (DM) is the result of insulin deficiency or insensitivity and shows the multiple complications in diabetic patients [9]. Although 90–95% of DM patients are insulin-independent type (type 2 diabetes, T2D), the pancreatic islet undergoes a malfunction or dysfunction that results in a loss of an insulin-secretion ability during disease progression and leads to a need to replenish insulin [10]. The pathogenesis of T2D is insulin resistance, where the body has a low reaction to insulin even though insulin levels are normal or higher. The improvement of insulin resistance seeks to develop therapeutic access to remedy T2D. Among insulin sensitization drugs, curcumin and rutin have emerged as potential hypoglycemic compounds derived from nature. Curcumin is a natural phenol produced by the *Curcuma longa* (Zingiberaceae) plant which has been reported to adequately sensitize insulin by inhibiting ATPase activity for mitochondria [11]. Rutin is a flavonoid found in a large variety of plants, including *Carpobrotus edulis* (Aizoaceae) and *Ruta graveolens* (Rutaceae). These plants have been found to play a critical role in sensitizing insulin by downregulating the expression level of the protein-tyrosine phosphatase 1B [12]. One vital problem in insulin therapy is the paucity of patient consent. Therefore, TDD is a preferred route for diabetic therapy and is convenient for patients who cannot take medicine [13] or suffer from an invasive cure. However, the delivery reagents of TDD for insulin or insulin sensitizers (curcumin, rutin) while applying the problem to diabetic patients remain to be explored.

*Dendrocnide meyeniana* (Dm, Urticaceae) and *Urtica thunbergiana* Sieb. and Zucc (Ut, Urticaceae) are species with nettle stings at leaf blades formed by cells with high Si/Ca ratio [14,15]. Histamine, oxalic acid, and tartaric acid could be found in these stinging nettles. They play different roles in inducing contact dermatitis [16,17]. *Alocasia odora* (Lodd.) Spach (Ao, Araceae) is a traditional herbal medicine for treating abdominal pain, cholera, hernias, stomach aches, heals wounds, and fungi [18,19]. The juice from the Ao leaf, which contains insoluble calcium oxalate, saponin, and alocasin, was also found to cause contact dermatitis [20,21]. From the cumulative information, Dm, Ut, and Ao could induce contact dermatitis through contact with leaf and juice from these plants. However, the conceivable potential of those sources remains to be explored, particularly in the permeation enhancer uses for TDD in the context of diabetes treatment. In this study, for the first time, the enhancement effect of Dm, Ut, and Ao on the in vivo permeation of insulin or insulin sensitizers (curcumin and rutin) through diet-induced diabetic mouse skin has been investigated. The adverse effect of Dm, Ut, and Ao used as a permeation enhancer in TDD was also examined through in vitro cell analysis and in vivo tests. In this work, capsaicin (Ca) isolated from *Capsicum annuum* (Solanaceae) was used as a positive control for the penetration enhancer since it has been shown to have analgesic advantages in post-therapeutic neuralgia, painful polyneuropathies, and surgical neuropathic syndromes [22]. Furthermore, the capsaicin 8% patch has been approved by the Food and Drug Administration (FDA) for post-therapeutic neuralgia [23].

## 2. Materials and Methods

### 2.1. Chemicals and Reagents

General chemicals used in this study, including rutin (R5143), curcumin (C1386), capsaicin (M2028), DMSO (D2650), trypan blue (TB, T8154), and 3-(4, 5-dimethylthiazol-2-yl)-2, 5-diphenyltetrazolium bromide (MTT, M5655), were purchased from Sigma-Aldrich (St. Louis, MO, USA). Reagents and mediums for cell culture were obtained from Thermo Fisher Scientific (Waltham, MA, USA). Hand cream (the compositions including water, glycerin, cetearyl alcohol, stearic acid, sodium cetearyl sulfate, methylparaben, propylparaben, dilauryl thiodipropionate, and sodium sulfate) was purchased from Neutrogena (Johnson and Johnson, New Brunswick, NJ, USA).

### 2.2. Plant Materials and Preparation of Sting Crude Extracts

Collections of *Dendrocnide meyeniana* (Dm, NCBI Taxonomy ID: 1399671) and *Urtica thunbergiana* Sieb. and Zucc (Ut, NCBI Taxonomy ID: 1303703), and *Alocasia odora* (Lodd.) Spach (Ao, NCBI Taxonomy ID: 174188) were from different localities: Dm from the riverside of the Mugua River (Shoufeng, Hualien, Taiwan), Ut from the lakeside of Liyu Lake (Shoufeng, Hualien, Taiwan), and Ao from the National Dong Hwa University (NDHU) (Shoufeng, Hualien, Taiwan). After collections, all plants were identified according to their morphology and their features were used to confirm species. Fresh leaves of Dm, Ut, and Ao were dried at room temperature (RT), cut into pieces, ground using a mill, sieved through a no. 40 mesh, and extracted by soaking with 50% ethanol/H_2_O at RT for one week. The extracts of Dm, Ut, and Ao were collected by filtration and vacuum evaporation. All extracts were stored at 4 °C and avoided exposure to light. In all performed experiments, the crude extracts and rutin were dissolved in the distilled water while curcumin was dissolved in ethanol. Capsaicin was solubilized in 10% Tween 80, 10% ethanol, and 0.9% saline. The prepared samples were filtered by sterile syringe filter with a 0.45 µm pore size. A voucher specimen (No. 106012-1) was deposited in the Department of Life Science and Institute of Biotechnology, National Dong Hwa University, Hualien, Taiwan.

### 2.3. Cell Culture

Human skin keratinocyte cells HaCaT were obtained from the Bioresource Collection and Research Center (BCRC, Hsinchu, Taiwan). Cells were grown in high-glucose Dulbecco’s modified eagle’s medium with 10% fetal bovine serum (FBS) and 1% antibiotic-antimycotic supplementation under culture condition (37 °C, 5% CO_2_). All experiments were carried out within 10 passes for consistency and accuracy.

### 2.4. Cell Viability Assay

The cell viability test was carried out in two different methods: TB exclusion and MTT assays. HaCaT cells were seeded at 5 × 10^4^ cells per well in 24-well plates and cultured overnight for adherence. Cells were treated with 20, 50, and 100 µg/mL of Dm, Ut, or Ao, respectively, for 30, 60, 90, and 120 min. For TB staining, cell suspension was diluted with 0.4% TB at a ratio 1:1 and further counted using hemocytometer (Neubauer, Germany). For the MTT experiment, cells were added with 30 μL/well MTT solution (25 μg/mL) and incubated for additional 4 h. Then the medium was removed, formazan was solubilized by the addition of 100 μL/well dimethyl sulfoxide (DMSO), and OD value was measured at 570 nm using α-screen multiplate reader (Perkin-Elmer, Waltham, MA, USA). The percentage of viable cells was determined by comparing with an untreated control.

### 2.5. Animal Source and Care

BALB/c male mice (8 weeks old) and ICR male mice (6 weeks old) were obtained from BioLASCO (Taipei, Taiwan) and housed in a pathogen-free environment under a constant room temperature of 25 °C, humidity of 50 ± 5%, and on a 12 h dark/light cycle. Mice had free access to water and food during the study period. All procedures were carried out in accordance with the “Guide for the Care and Use of Laboratory Animals” of NDHU and approved by the NDHU Animal Ethics Committee (Relevant permit number: NSCB04-001).

### 2.6. Diet-Induced Obesity (DIO) Model

After one week of adaptation, ICR mice were fed with a high-fat diet (150 g lard/kg commercial diet) and 60% (*v*/*v*) fructose for 14 weeks. At week 10, ICR mice fasted for 10 h prior to oral gavage 4 g/kg body weight of D-glucose. Blood from the tail vein was collected, and then the blood sugar level was analyzed using an EasyTouch^®^ GCU glucose analyzer (Bioptik Technology, Miaoli, Taiwan) for 120 min every 30 min. When blood sugar levels were higher than 160 mg/dL at 120 min postprandial, mice were defined as glucose intolerant.

### 2.7. Insulin-Glucose Tolerance Test (IGT)

Glucose intolerant mice were intraperitoneally (i.p.) injected with 4.25 U/kg body weight of insulin solution (Novo Nordisk, Kalundborg, Denmark). If the blood sugar level was higher than 126 mg/dL at 120 min after insulin injection, mice were considered as diabetic.

### 2.8. Oral Glucose Tolerance Test (OGTT)

Diabetes-induced ICR7 mice were shaved on the neck and back followed by fasting for 10 h. After determining preprandial blood sugar level, diabetes-induced ICR7 mice were smeared with Dm (2 mg/mL), Ut (2 mg/mL), Ao (2 mg/mL), or Ca (0.25 mg/mL) with/without insulin (10 pM; 0.17 U), curcumin (20 μg/mL), or rutin (80 μg/mL), respectively; followed by the oral gavage of glucose. Blood from the tail vein was collected, and then the blood sugar level was analyzed by glucose analyzer for 120 min at an interval of every 30 min. The area under the curve for blood sugar (AUC) was calculated using the trapezoidal method [24].

### 2.9. Skin Irritation Test

BALB/c mice were shaved on their neck and back, and then smeared with 100 µL of Dm (2 mg/mL), Ut (2 mg/mL), or Ao (2 mg/mL), respectively. The skin was then examined for erythema/edema based on Draize dermal irritation scoring system [25], and treating areas were snapped once per hour for 4 h.

### 2.10. Histological Analysis of Skin Sections

Diabetes-induced ICR mice were shaven on their neck and back and smeared with 100 µL of Dm (2 mg/mL), Ut (2 mg/mL), or Ao (2 mg/mL) together with hand cream (20 mg, Neutrogena, Seoul, Korea), respectively. After 30 min, mice skin was cut and fixed in 4% paraformaldehyde (PFA) solution. The fixed skin was embedded by paraffin followed by hematoxylin and eosin (H&E) staining according to standard procedures. Images were acquired using a Nikon digital camera (Sight DS-U1, Tokyo, Japan).

### 2.11. Statistical Analysis

Data were shown as mean ± standard deviation (SD) of three independent experiments. A *p*-value less than 0.05 was considered to be statistically significantly different (* *p* < 0.05) using one-way ANOVA with Dunnett’s test.

## 3. Results

### 3.1. Effect of Sting Crude Extracts Dm, Ut, and Ao on In Vivo Skin Irritation

When the skin contacts the natural crude extracts, it is strikingly crucial to fortify so that Dm, Ut, and Ao will not cause serious contact sensitization responses. For this reason, a skin-irritation test was performed on mice by smearing mouse skin with 100 µL of Dm, Ut, or Ao at a concentration of 2 mg/mL, respectively. The outcome has been graded depending upon the degree of erythema caused as follows: 0—no erythema, 1—very slight erythema, 2—well-defined erythema, 3—moderate to severe erythema, and 4—severe erythema [25]. Table 1 shows that Dm, Ut, and Ao caused slight erythema when tested on the mouse skin. Moreover, no edema was observed in the mouse skin treated with three individual extracts. The observations of the mouse skin after 1 h and 4 h of Dm (Figure 1A), Ut (Figure 1B), and Ao (Figure 1C) treatments were also associated with the dermal irritation scores. No signs of skin irritation were found after 6 h of the extract treatments (data not shown). Taken together, the sting crude extracts Dm, Ut, and Ao could induce barely perceptible signs of erythema without causing edema in the mouse skin.

*Dendrocnide meyeniana* (Dm), *Urtica thunbergiana* Sieb. and Zucc (Ut), and *Alocasia odora* (Lodd.) Spach (Ao) are tested in mouse skin. 0—no erythema, 1—very slight erythema, 2—well-defined erythema, 3—moderate to severe erythema, and 4—severe erythema [25].

### 3.2. Effect of Sting Crude Extracts Dm, Ut, and Ao on Mouse Skin Epidermis

The elevation of stratum corneum (SC) permeability plays a key role in increasing transdermal drug delivery. To investigate the effect of the sting crude extracts Dm, Ut, and Ao on the structure of the mouse skin, diabetes-induced ICR mice were exposed to 100 µL of Dm (2 mg/mL), Ut (2 mg/mL), or Ao (2 mg/mL) mixed with hand cream (20 mg) for 30 min and then examined histologically to image the appearance of the mouse skin. Figure 2A,B show representative pictures of the intact mouse skin epidermis without any damage when left untreated (Figure 2A) or only smeared with hand cream (Figure 2B). On the contrary, treatment of the mouse skin with Dm (Figure 2C), Ut (Figure 2D), or Ao (Figure 2E) plus hand cream resulted in structural damage of SC as well as viable epidermis (stratum granulosum, stratum spinosum), suggesting that the sting crude extracts Dm, Ut, and Ao may be potential penetration enhancers for increasing the uptake of medicines into the skin.

### 3.3. Effect of Sting Crude Extracts Dm, Ut, and Ao on Human Skin Keratinocytes

Skin irritation and cell damage are among the most important standards to evaluate whether or not the penetration enhancers could be approved for clinical use in spite of their adequately sufficient fulfilment in enhancing the permeation of drug molecules. Hence, the cell viability evaluation of the sting crude extracts was studied using trypan blue and MTT assays. Figure 3 shows that the data of human skin keratinocytes HaCaT incubated with various concentrations of extracts. All tested crude extracts exhibited much less cytotoxicity (<3%) at 2 mg/mL for 120 min incubation in the HaCaT cells in both trypan blue exclusion (Figure 3A) and MTT staining (Figure 3B). Furthermore, in a longer incubation period (48 h), the sting crude extracts Dm, Ut, and Ao displayed a similar extent of the HaCaT cell cytotoxicity as compared with untreated cells (data not shown). Additionally, no remarkable change in cell morphology was observed in the HaCaT untreated cells and cells treated with Dm, Ut, or Ao, respectively at 2 mg/mL for 120 min. Inverted microscope images showed that HaCaT cells were cubical in shape with high cell-to-cell packing in both untreated and treated groups (Figure 3C).

### 3.4. Effect of Sting Crude Extracts Dm, Ut, and Ao on In Vivo Skin Permeation of Insulin and Insulin Sensitizers

To investigate the effect of the sting crude extracts as the enhancers on the in vivo skin permeation of the insulin and insulin sensitizers, diabetes-induced ICR7 mice were smeared with Dm, Ut, or Ao at 2 mg/mL or Ca at 0.25 mg/mL with/without insulin (10 pM; 0.17 U), curcumin (20 μg/mL), or rutin (80 μg/mL), respectively; followed by the oral gavage of glucose and blood sampling. The results showed that applying individual crude extracts Dm, Ut, or Ao or the capsaicin compound to the mouse skin were found to have no effect on the reduction of the blood sugar level (Figure 4A) as revealed by insignificant increases of AUC for glucose in the crude extracts pretreated skin as compared with the untreated group (Figure 4B, *p* > 0.05). When checking the positive control, intraperitoneal injection of insulin (10 pM) produced a significant decrease in serum glucose (Figure 4A,B, *p* < 0.05). Skin permeation of the insulin without the enhancers (direct smear insulin) was negligibly proven by the elevated blood sugar level with similar concentrations as compared with untreated skin. However, the addition of the crude extracts or capsaicin as the enhancers increased the skin permeation of the insulin as compared with the control groups (untreated skin or insulin treated without enhancers) demonstrated by the strongly decreasing blood sugar levels (Figure 4C,D, *p* < 0.05).

Curcumin and rutin are well-known insulin sensitizers that improve insulin resistance and lower blood sugar levels by oral administration [26,27]. Figure 4E,H show the permeation profiles of curcumin (Figure 4E,F) and rutin (Figure 4G,H) from the various formulations. However, skin permeation of curcumin (20 μg/mL) or rutin (80 μg/mL) significantly raised the blood sugar level as compared with oral curcumin (0.6 mg/kg body weight) or rutin (2.5 mg/kg body weight) (*p* < 0.05). This indicates that curcumin and rutin cannot easily penetrate the intact skin; whereas, the transdermal treatments of curcumin or rutin with induvial enhancer Dm, Ut, or Ao, respectively, were more efficient than transdermal treatments of curcumin or rutin without the enhancers in reducing blood sugar concentration (*p* < 0.05).

Taken together, smearing with the sting crude extracts Dm, Ut, or Ao was found to cause increases in skin penetration of the insulin and insulin sensitizers (curcumin and rutin), which, consequently, affected the blood sugar level to meet non-invasion demands.

## 4. Discussion

For decades, the use of skin as a target for drug transfer has been a fascinating preference to traditional approaches including intravenous and oral administration. However, human skin has a superior permeability barrier to prevent entry of nearly all but small molecules [28]. Therefore, exploration for the optimal skin penetration enhancer has been the spotlight of appreciable investigation throughout the years. Herein, for the first time, we discovered that the natural sting crude extracts of Dm, Ut, and Ao impaired the structures of stratum corneum, stratum granulosum, and stratum spinosum of mouse skin epidermis (Figure 2) to boost the skin permeation of the insulin and insulin sensitizers (curcumin and rutin) for reducing the blood glucose level in diabetic ICR mice (Figure 4) while causing slight skin erythema without skin edema (Figure 1) or any damage to the HaCaT human skin keratinocytes (Figure 3).

It has been shown that the stings of Dm are found to cause the acute dermatitis by its component formic acid via contact [29]. The formic acid can induce the dilatation of blood vessels, which allows calcium to enter into cells [30]. Major constituents of Ut nettle stings contain histamine, formic acid, and serotonin that elicit slight erythema in the first 10 min of treatment and induce less pain sensation [15,16,17,31,32]. In addition, Ao which contains calcium oxalate is found to cause slight skin stimulation and its methanol extract has been shown to have antifungal bioactivity [19]. As an ideal penetration enhancer, the molecule should be nontoxic and nonirritating [33]. In this study, we showed that three sting extracts Dm, Ut, and Ao led to no or minimal perceptible erythema without edema on the mouse skin (Figure 1). Furthermore, the extracts had no damage on the cell viability of the HaCaT human skin keratinocytes demonstrated by the MTT assay and Trypan blue exclusion, which excludes non-specific intracellular reduction of the tetrazolium that led to underestimating the results of cytotoxicity by MTT assay (Figure 3).

The skin, especially the SC layer, provides a serious barrier to drug permeation and restricts percutaneous bioavailability. Previous studies have shown that the skin permeability was further increased by an order of magnitude of 1 or 2 upon removal of the full epidermis as compared with removal of SC only [34]. Our data demonstrated that the sting crude extracts of Dm, Ut, and Ao influenced the structure of SC and also affected the SG and SS of viable epidermis (Figure 2).

The administration route of the insulin via injection is a common approach to reduce blood glucose for diabetic patients. However, this needle-based method requires remarkable training and assiduous care by patients, which usually leads to a poor impact [35]. Transdermal delivery of the insulin is appealing as a noninvasive route that provides the usefulness of a transdermal patch, including the use of liposomes [36] and ultrasound [37]. Our results showed that 10 pM of insulin immersed with 2 mg/mL of Dm, Ut, and Ao, or with 0.25 mg/mL of Ca could significantly reduce blood sugar levels via transdermal transfusion as compared with insulin TDD as a negative control (*p* < 0.05, Figure 4C,D), suggesting that insulin can pass through the skin and enter into blood vessel with the direct support of the penetration enhancers.

Diabetes could increase plasma free fatty acid, cholesterol, and triglyceride concentrations and cause stroke or myocardial infarction [38]. As we know, curcumin has been reported to reduce blood glucose via inhibiting IL-6, MCP-1, TNF-α, and NF-κB [39,40], enhance GLP-1 secretion to reduce the dose of diabetic medicines [41], and increase the hepatic glycogen and skeletal muscle lipoprotein lipase [38]. In this study, curcumin combined with Dm, Ut, or Ao strongly decreased blood sugar levels by the transdermal delivery as compared with curcumin TDD as a negative control (*p* < 0.05, Figure 4E,F).

Rutin also inhibited the generation of advanced glycation end products (AGEs), and improved liver protection [42]. Previous reports have shown that intensive glycemic control can reduce the aggravation of diabetic symptoms [43]. Moreover, glucose transporter 4 (GLUT4) was induced by the PI3K and Akt/PKB pathways, and effectively control glucose regulator [44,45]. Rutin could promote the PI3K and Akt/PKB pathways, and also potentiate GLUT4 to control glucose level [26]. Our results demonstrated that rutin plus Dm, Ut, or Ao could be employed to effectively reduce blood glucose level by the transdermal diffusion as compared with rutin TDD as a negative control (*p* < 0.05; Figure 4G,H).

## 5. Conclusions

Altogether, the data of the present investigation show that the sting plant extracts Dm, Ut, and Ao as the penetration enhancers may alter the skin epidermis morphology to facilitate passive diffusion of insulin, curcumin, and rutin into the skin and blood vessel to efficiently moderate the blood sugar levels. Moreover, based on our findings, diabetes-induced mice are found to be an appropriate model organism used for imitating human diabetes for investigating whether the transdermal drug delivery is feasible and worth using as a diabetic remedy per se. Lastly, the application of the sting plants as a permeation agent for dermal medicine delivery may provide a new clue and an alternative approach as a non-invasion route for drug administration.

## Figures and Tables

**Figure 1 medicina-55-00121-f001:**
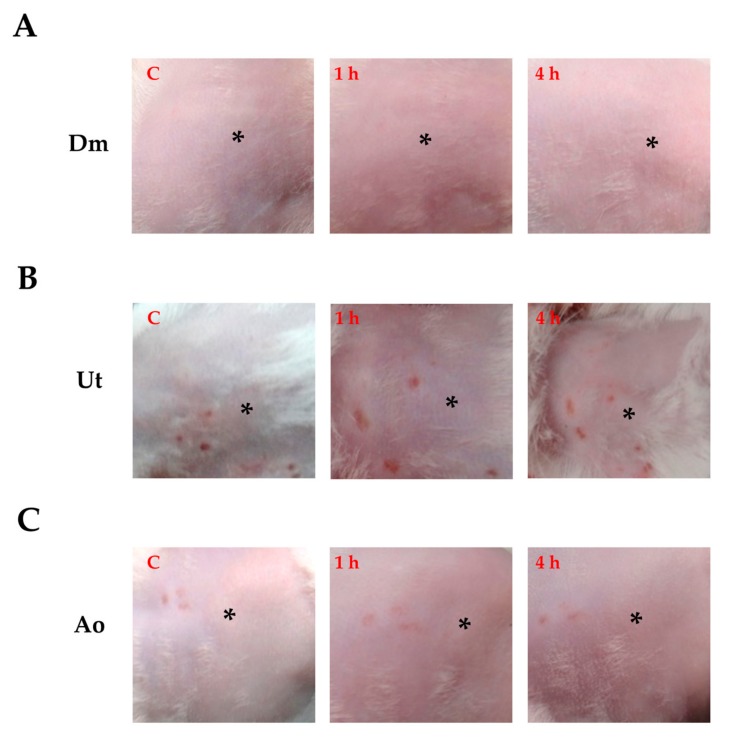
Observation in the color changes of the mouse skin in vivo after smearing with (**A**) Dm (2 mg/mL in H_2_O), (**B**) Ut (2 mg/mL in H_2_O), and (**C**) Ao (2 mg/mL in H_2_O) at 1 h and 4 h. The asterisk (*) showed changes in mouse skin redness; C: untreated mouse skin as control. Data are representative of three independent experiments.

**Figure 2 medicina-55-00121-f002:**
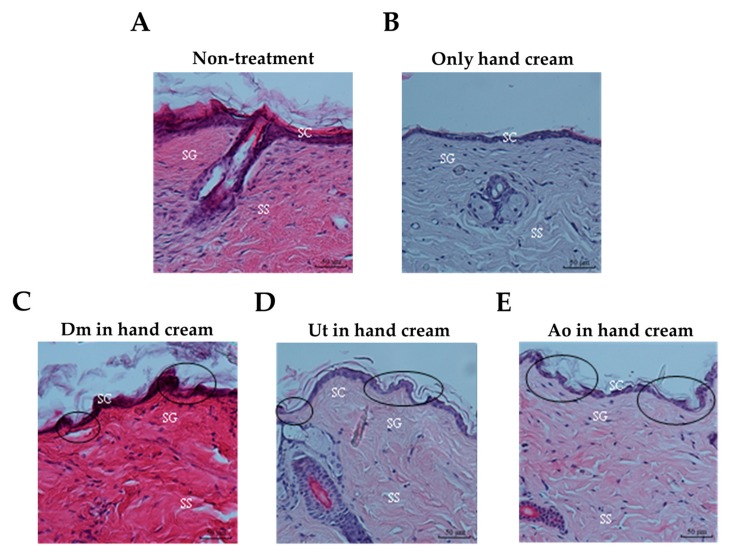
Changes in mouse skin morphology at 30 min after treatment with various reagents. H&E stained section of the mouse skin epidermis and dermis of (**A**) non-treatment, (**B**) only hand cream (20 mg), (**C**) Ut (2 mg/mL in H_2_O) plus hand cream (20 mg), (**D**) Dm (2 mg/mL in H_2_O) plus hand cream (20 mg), and (**E**) Ao (2 mg/mL in H_2_O) plus hand cream (20 mg). Circles show the damage areas of the mouse skin. SC: stratum corneum, SG: stratum granulosum, and SS: stratum spinosum. Data are representative of three independent experiments. Magnification, ×200.

**Figure 3 medicina-55-00121-f003:**
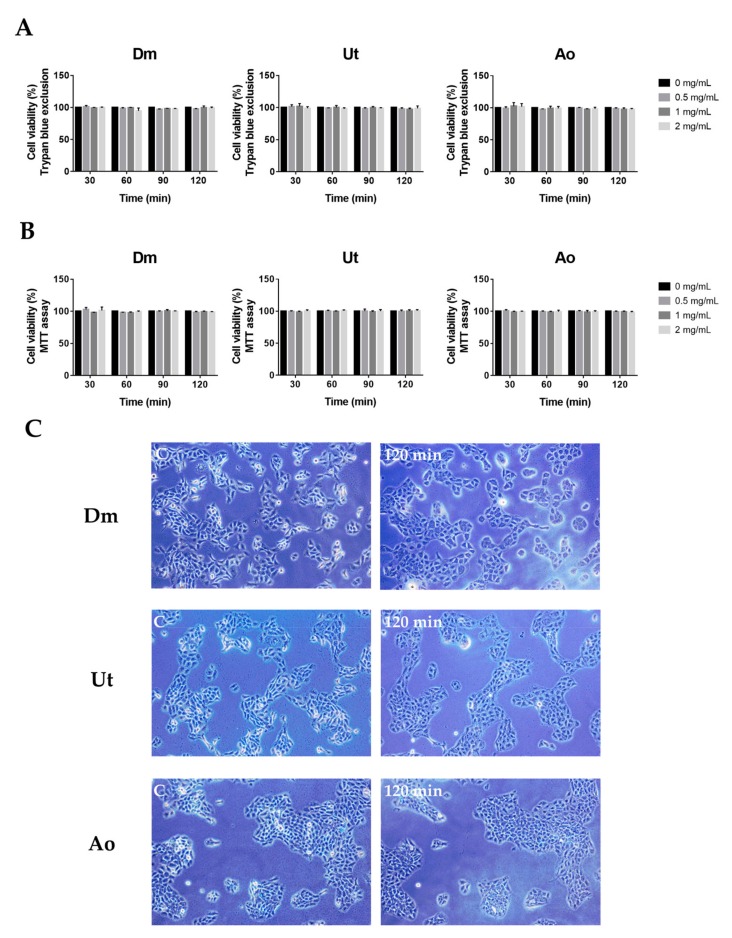
Cytotoxic effect of three sting crude extracts against keratinocytes. HaCaT cells were treated with various concentrations of the sting crude extracts at different time points. Cell viability was analyzed by (**A**) trypan blue exclusion and (**B**) MTT assay, respectively. The cell viability was determined and presented as a percentage of the untreated cells. Data are presented as mean ± SD in three independent experiments. (**C**) Representative inverted microscope images of the HaCaT cells, which were untreated as control—C, or treated with crude extracts for 120 min. Magnification, ×100.

**Figure 4 medicina-55-00121-f004:**
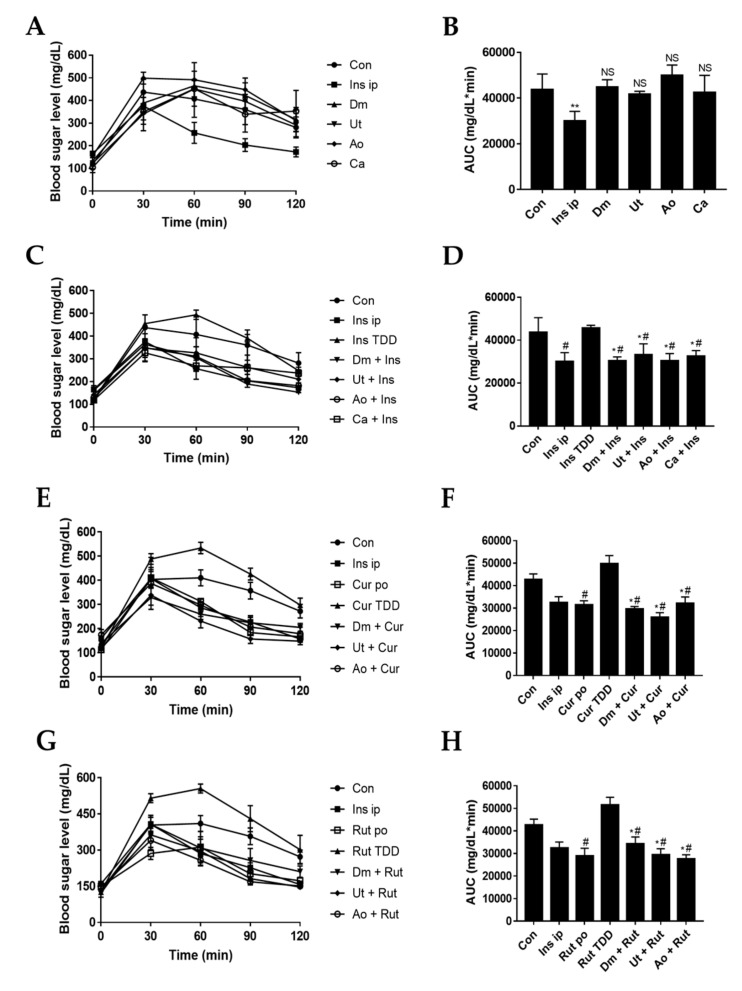
Hypoglycemic efficacy of insulin or insulin sensitizers combined with the sting crude extracts. Mice were fasted for 10 h, then preprandial blood glucose level was determined (0 min). Mouse skin was smeared with Dm (2 mg/mL), Ut (2 mg/mL), Ao (2 mg/mL), or Ca (0.25 mg/mL) with/without insulin (10 pM; 0.17 U), curcumin (20 μg/mL), or rutin (80 μg/mL), respectively. Mice received the oral gavage of glucose, then blood sugar level and area under curve (AUC) were checked with (**A**,**B**) only crude extracts, (**C**,**D**) crude extracts plus insulin, (**E**,**F**) crude extracts plus curcumin, and (**G**,**H**) crude extracts plus rutin every 30 min until 120 min. Ins ip: intraperitoneal injection of insulin, Ins TDD: transdermal drug delivery of insulin, Cur po: oral administration of curcumin, Cur TDD: transdermal drug delivery of curcumin, Rut po: oral administration of rutin, Rut TDD: transdermal drug delivery of rutin, Con: untreated mice as control, NS: not significant vs. control, * *p* < 0.05 vs. control, # *p* < 0.05 vs. insulin TDD, curcumin TDD or rutin TDD.

**Table 1 medicina-55-00121-t001:** Skin irritation effect of sting crude extracts.

Penetration Enhancer	Score of Skin Irritation
Dm (2 mg/mL)	1
Ut (2 mg/mL)	1
Ao (2 mg/mL)	1

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
