# Peer review of "Effects of Sting Plant Extracts as Penetration Enhancers on Transdermal Delivery of Hypoglycemic Compounds"

_medicina, 2019, doi:10.3390/medicina55050121_

Round 1
Reviewer 1 Report
The manuscript " Sting Plant Extracts as Penetration Enhancers Potentiate Transdermal Delivery of Hypoglycemic Compounds" submitted for publication in the medicina using three different Chinese herbal extracts as the adjuvants in transdermal delivery of hyperglycemic compounds.
The manuscript is well written and the study is well design. However, there is little concern about the overall study prior to be accepted for publication. As author mentioned that the extracts able to potentiate the delivery of hyperglycemic compounds. Would like to make it clear about which compounds that the compounds are rutin and curcumin? Why particularly rutin and curcumin? Any previous in-vitro/in-vivo data showed that there are the potential hyperglycemic compounds? It would be interesting to know how the extracts able to act as the adjuvants. It should be explaining further? Are those three extracts used together with the equal ratio or individually? 3 hours used to access the cytotoxicity of the extracts is too short, though it has been mentioned that it is safe to use and non-toxic with data not shown. It would be great to show the chemical profile of all the extracts with some explanations about what are the chemical constituents within it.
Author Response
Manuscript ID: Medicina-468412
We greatly appreciate the helpful comments of the reviewers. We have made the necessary changes, which we believe substantially improved the manuscript. In response to the reviewers’ comments, we have added new sentences and edited existing sentences in the revised version according the comments. All of the changes are yellow highlighted in the revised manuscript. Our point-by-point responses are set out below.
Responses to specific comments from Reviewer #1
The general comment of Reviewer #1:
The manuscript "Sting Plant Extracts as Penetration Enhancers Potentiate Transdermal Delivery of Hypoglycemic Compounds" submitted for publication in the Medicina using three different Chinese herbal extracts as the adjuvants in transdermal delivery of hyperglycemic compounds.
The manuscript is well written and the study is well design. However, there is little concern about the overall study prior to be accepted for publication. As authors mentioned that the extracts able to potentiate the delivery of hyperglycemic compounds.
Specific comments:
Reviewer comment 1: Would like to make it clear about which compounds that the compounds are rutin and curcumin?
Response: The authors thank the eminent reviewer’s comment.
We purchased Rutin (Catalog number R5143, Sigma Aldrich, USA) and Curcumin (Catalog number C1386, Sigma Aldrich, USA).
To correct this, we have shown how we got them in the “Chemicals and Reagents” section of the Materials and Methods (page 2, line 88-90 of the revised manuscript):
“General chemicals used in this study including rutin (R5143), curcumin (C1386), capsaicin (M2028), DMSO (D2650), trypan blue (TB, T8154), and 3-(4, 5-dimethylthiazol-2-yl)-2, 5-diphenyltetrazolium bromide (MTT, M5655) were purchased from Sigma-Aldrich (Missouri, USA).”
2) Reviewer comment 2: Why particularly rutin and curcumin?
Response: The authors thank the eminent reviewer’s comment.
We have added some information based on the above comment of Reviewer in the Introduction section (page 2, line 56-64 of the revised manuscript):
“The pathogenesis of T2DM is insulin resistance, in which body has a low reaction to insulin even though insulin stands at the normal or even higher level in the blood. Improvement of insulin resistance has been a therapeutic access in the remedy of T2DM. Among insulin sensitization drugs, curcumin and rutin emerge as the potential hypoglycemic compounds derived from nature. Curcumin is a natural phenol produced by Curcuma longa (Zingiberaceae) plant which has been reported to sensitize insulin adequately by inhibiting ATPase activity of mitochondria [11]. Rutin is a flavonoid found in a large variety of plants including Carpobrotus edulis (Aizoaceae) and Ruta graveolens (Rutaceae) which played a critical role in sensitizing insulin by down-regulating the expression level of protein-tyrosine phosphatase 1B [12].”
We have also corrected the References (page 11-12, line 367-372 of the revised manuscript) as follow:
“11. Hong, S.; Pedersen, P.L. ATP synthase and the actions of inhibitors utilized to study its roles in human health, disease, and other scientific areas. Microbiology and molecular biology reviews : MMBR 2008, 72, 590-641, Table of Contents, doi:10.1128/MMBR.00016-08.
12. Lee, D.G.; Jang, I.S.; Yang, K.E.; Yoon, S.J.; Baek, S.; Lee, J.Y.; Suzuki, T.; Chung, K.Y.; Woo, S.H.; Choi, J.S. Effect of rutin from tartary buckwheat sprout on serum glucose-lowering in animal model of type 2 diabetes. Acta pharmaceutica 2016, 66, 297-302, doi:10.1515/acph-2016-0021.”
Two more citations were added and all references were renumbered in revised manuscript.
3) Reviewer comment 3: Any previous in-vitro/in-vivo data showed that there are the potential hyperglycemic compounds?
Response: The authors thank the eminent reviewer’s comment.
To the best of our knowledge, there is no potential hyperglycemic compound discovered so far.
4) Reviewer comment 4: It would be interesting to know how the extracts able to act as the adjuvants. It should be explaining further?
Response: The authors thank the eminent reviewer’s comment.
Skin is one of the primary challenges to deliver a majority of the drugs through transdermal route. The immense hindrance for TDD is found in the stratum corneum (SC) of skin that reduces water evaporation from dermis and blocks the permeation of chemical and biological drugs.
The plants used in the study are species with nettle stings at leaf blades. The plants contain some chemicals including histamine, oxalic acid, and tartaric acid playing different roles in inducing contact dermatitis.
We hypothesized that the crude extracts of three plants can disrupt the complex structure of SC allowing drug like insulin can be used as TDD to penetrate the damaged SC structure. The results confirmed our supposition based on the in vivo OGTT test. That was the reason why we concluded the extracts can play a role as the adjuvants in TDD.
In the submitted manuscript, we have described the reason why crude extracts can be the potential adjuvants in the Discussion (page 9, line 275-284 of the submitted manuscript) as followings:
“For decades, the use of skin as a mean for drug transfer has been a fascinating preference to traditional approaches including intravenous and oral administration. However, human skin is arranged to be a superior permeability barrier to prevent entry of nearly all but small molecules [19]. So the exploration for the optimal skin penetration enhancer has been the spotlight of appreciable investigation exertion throughout the years. Here, for the first time, we discovered that the natural sting crude extracts of Dm, Ut, and Ao impaired the structures of stratum corneum, stratum granulosum, and stratum spinosum of the mouse skin epidermis (Figure 2) to boost the skin permeation of insulin and insulin sensitizers (curcumin and rutin) for reducing the blood sugar level in diabetic ICR mice (Figure 4) while causing very slight skin erythema without skin edema (Figure 1) or any damage to HaCaT human skin keratinocytes (Figure 3).”
5) Reviewer comment 5: Are those three extracts used together with the equal ratio or individually?
Response: The authors thank the eminent reviewer’s comment.
Three extracts were treated individually.
To prevent misunderstanding, we have edited in the “Cell viability assay” section of the Materials and Methods (page 3, line 119-120 of the revised manuscript):
“Cells were treated with 20, 50, and 100 µg/mL of Dm, Ut, or Ao, respectively for 30, 60, 90, and 120 min.”
6) Reviewer comment 6: 2 hours used to access the cytotoxicity of the extracts is too short, though it has been mentioned that it is safe to use and non-toxic with data not shown.
Response: The authors thank the eminent reviewer’s comment.
In the OGTT test, we smeared diabetic mice with the extracts for 120 min. That was why we tested the cytotoxicity of the extracts on HaCaT cell line for 120 min. We would like to know whether the extracts can induce any damage to the skin by in vitro system.
However, we have also checked its cytotoxicity in HaCaT cells by MTT assay for 48 hr and the data are not shown in this submitted manuscript.
We have mentioned about that information in the section 3.3 of the Results (page 6, line 217-219 of the submitted manuscript) as follow:
“Furthermore, in a longer incubation period (48 h), sting crude extracts Dm, Ut, and Ao displayed a similar extent of HaCaT cytotoxicity as compared with untreated cells (data not shown).”
7) Reviewer comment 7: It would be great to show the chemical profile of all the extracts with some explanations about what are the chemical constituents within it.
Response: The authors thank the eminent reviewer’s comment.
Previous studies have shown some data related to the chemical constituents of Dm and Ut. We have cited those works as our references in the Introduction section (page 2, line 69-72 and References (page 11, line 375-385) of the submitted manuscript:
Introduction
“Dendrocnide meyeniana (Dm, Urticaceae) and Urtica thunbergiana Sieb. & Zucc (Ut, Urticaceae) are species with nettle stings at leaf blades formed by cells with high Si/Ca ratio [14,15]. Histamine, oxalic acid, and tartaric acid could be found in these stinging nettles and play different roles in inducing contact dermatitis [16,17].”
References
“14. Fu, H.-Y.; Chen, S.-J.; Kuo-Huang, L.-L. Comparative study on the stinging trichomes and some related epidermal structures in the leaves of Dendrocnide meyeniana, Girardinia diversifolia, and Urtica thunbergiana. Taiwania 2003, 48, 213-223.
15. Oliver, F.; Amon, E.; Breathnach, A.; Francis, D.; Sarathchandra, P.; Kobza Black, A.; Greaves, M. Contact urticaria due to the common stinging nettle (Urtica dioica)—histological, ultrastructural and pharmacological studies. Clinical and experimental dermatology 1991, 16, 1-7.
16. Taskila, K.; Saarinen, J.V.; Harvima, I.T.; Harvima, R.J. Histamine and LTC4 in stinging nettle-induced urticaria. Allergy 2000, 55, 680-681.
17. Fu, H.Y.; Chen, S.J.; Chen, R.F.; Ding, W.H.; Kuo-Huang, L.L.; Huang, R.N. Identification of oxalic acid and tartaric acid as major persistent pain-inducing toxins in the stinging hairs of the nettle, Urtica thunbergiana. Ann Bot 2006, 98, 57-65.”
We have found some new references mentioning about the chemical constituents of Ao.
We have now described the chemical constituents of Ao in the Introduction (page 2, line 72-75 of the revised manuscript):
“Alocasia odora (Lodd.) Spach (Ao, Araceae) is a traditional herbal medicine for treating abdominal pain, cholera, hernia, stomach ache, wound healing, and fungi [18,19]. The juice from Ao leaf containing insoluble calcium oxalate, saponin, and alocasin also caused contact dermatitis [20,21].”
We have also corrected the References (page 12, line 386-393 of the revised manuscript) as follow:
“18. Viet, L.; Houghton, P.; Forbes, B.; Corcoran, O.; Hylands, P. Wound healing activity of Alocasia odora (Roxb.) Koch. Planta Medica 2006, 72, 024.
19. Wang, Y.-D.; Liao, M.-D.; Zhang, J.-L.; XU, H.-H. Tested on antifungal activity of Alocasia odora methanol extract. PESTICIDES-SHENYANG- 2006, 45, 744.
20. Moon, J.M.; Lee, B.K.; Chun, B.J. Toxicities of raw Alocasia odora. Hum. Exp. Toxicol. 2011, 30, 1720-1723, doi:10.1177/0960327110393760.
21. Chan, T.; Chan, L.; Tam, L.; Critchley, J. Neurotoxicity following the ingestion of a Chinese medicinal plant, Alocasia macrorrhiza. Human & experimental toxicology 1995, 14, 727-728.”
Three more citations were added and all references were renumbered in revised manuscript.
Again, we greatly appreciate all of your helpful and constructive comments. We hope the eminent reviewer can accept our extensive response. We look forward to your reply in your convenient time.
Sincerely yours,
Ching-Feng Weng, Ph.D., Professor
Department of Life Science and Institute of Biotechnology
National Dong Hwa University, Shoufeng, Hualien 97401, Taiwan
Phone: +886-3-890-3637

Reviewer 2 Report
Manuscript Number: medicina-468412
Title: Sting Plant Extracts as Penetration Enhancers Potentiate Transdermal Delivery of Hypoglycemic Compounds
Comments to Authors
Recommendation: Publish with major changes
The article "Sting Plant Extracts as Penetration Enhancers Potentiate Transdermal Delivery of Hypoglycemic Compounds” has the goal to study the effect of herbal penetration enhancers on in vivo permeation of insulin and insulin sensitizers (curcumin and rutin) through diabetic-induced mouse skin.
Although the interesting theme, and that the design to determine and to confirm the expected pharmacology effects was set on the good base, some parts in experimental and discussion sections need improvement.
In Introduction section, authors should provide the information regarding synthesized compounds used as commercially available penetration enhancers, as the authors had the aim to reveal “…that natural plant extracts could be preferred over the chemically synthesized molecules as safe and potent penetration enhancers to stimulate the transdermal absorption of drugs”. In addition, please, provide the sufficient explanation and literature data regarding the investigation of curcumin and rutin as insulin sensitizers in TDD, to introduce the reader with the known literature regarding their up-to-now effects, and the reason why they were chosen for this kind of administration, as they were known for their per os application.
The most important and necessary addition to Introduction part is to make logical their choice of investigated plants for this particular indication - why in TDD for diabetic therapy?
When first mentioning the full Latin name (species name, author name and Family, as well), should be provided for the plant species mentioned in the text.
In M&M section, it is necessary to describe the whole procedure of obtaining the extracts for investigation, giving the dry residue, as the dose of administration further in the experiments were given as concentration. In all performed experiments, authors are asked to provide the solvent in which the crude extracts of the investigated plants were dissolved. Besides, the preparation of curcumin and rutin solutions that were administrated in the experiments should be explained, as the solubility of those compounds is different.
In Experimental section, authors are asked to give the composition of negative control - “hand cream”, as stated in text (line 176).
The main concern is the lack of positive control - as the enhancement effect of Ut, Dm and Ao extracts on in vivo insulin permeation through diet-induced diabetic mouse skin was investigated, the positive control should also be included in the experiment, meaning some of the synthetically obtained substances, for which clinically, or experimentally confirmed effects as enhancers of permeability in TDD exist in literature.
Page 8, line 230, please, correct ug/mL into μg/mL
Page 9, Figure 4, please, give the explanation for Rut op, Cur op in the legend of Figure 4.
Author Response
Manuscript ID: Medicina-468412
We greatly appreciate the helpful comments of the reviewers. We have made the necessary changes, which we believe substantially improved the manuscript. In response to the reviewers’ comments, we have added new sentences and edited existing sentences in the revised version according to the comments. All of the changes are yellow highlighted in the revised manuscript. Our point-by-point responses are set out below.
Responses to specific comments from Reviewer #2
The general comment of Reviewer #2:
The article "Sting Plant Extracts as Penetration Enhancers Potentiate Transdermal Delivery of Hypoglycemic Compounds” has the goal to study the effect of herbal penetration enhancers on in vivo permeation of insulin and insulin sensitizers (curcumin and rutin) through diabetic-induced mouse skin.
Although the interesting theme, and that the design to determine and to confirm the expected pharmacology effects was set on the good base, some parts in experimental and discussion sections need improvement.
Specific comments:
Reviewer comment 1: In Introduction section, authors should provide the information regarding synthesized compounds used as commercially available penetration enhancers, as the authors had the aim to reveal “…that natural plant extracts could be preferred over the chemically synthesized molecules as safe and potent penetration enhancers to stimulate the transdermal absorption of drugs”.
Response: The authors thank the eminent reviewer’s comment.
There are some chemically synthesized compounds as penetration enhancers causing severe skin irritation including DMSO, some fatty acids, and surfactants.
We have added the above information in the Introduction section (page 2, line 47-50 of the revised manuscript):
“For example, dimethyl sulphoxides (DMSO) is one of the most commonly studied PE which can cause erythema and wheal of SC [7] or oleic acid in propylene glycol caused severe skin irritation in the human volunteers [8].”
We have also corrected the References (page 11, line 359-362 of the revised manuscript) as follow:
“7. Kligman, A.M. Topical pharmacology and toxicology of dimethyl sulfoxide—Part 1. Jama 1965, 193, 796-804.
8. Loftsson, T.; Gildersleeve, N.; Bodor, N. The effect of vehicle additives on the transdermal delivery of nitroglycerin. Pharm Res-Dordr 1987, 4, 436-437.”
Two more citations were added and all references were renumbered in revised manuscript.
2) Reviewer comment 2: In addition, please, provide the sufficient explanation and literature data regarding the investigation of curcumin and rutin as insulin sensitizers in TDD, to introduce the reader with the known literature regarding their up-to-now effects, and the reason why they were chosen for this kind of administration, as they were known for their per os application.
Response: The authors thank the eminent reviewer’s comment.
We have added some information based on the above suggestions of Reviewer in the Introduction section (page 2, line 56-64 of the revised manuscript):
“The pathogenesis of T2DM is insulin resistance, in which body has a low reaction to insulin even though insulin stands at the normal or even higher level in the blood. Improvement of insulin resistance has been a therapeutic access in the remedy of T2DM. Among insulin sensitization drugs, curcumin and rutin emerge as the potential hypoglycemic compounds derived from nature. Curcumin is a natural phenol produced by Curcuma longa (Zingiberaceae) plant which has been reported to sensitize insulin adequately by inhibiting ATPase activity of mitochondria [11]. Rutin is a flavonoid found in a large variety of plants including Carpobrotus edulis (Aizoaceae) and Ruta graveolens (Rutaceae) which played a critical role in sensitizing insulin by down-regulating the expression level of protein-tyrosine phosphatase 1B [12].”
We have also corrected the References (page 11-12, line 367-372 of the revised manuscript) as follow:
“11. Hong, S.; Pedersen, P.L. ATP synthase and the actions of inhibitors utilized to study its roles in human health, disease, and other scientific areas. Microbiology and molecular biology reviews : MMBR 2008, 72, 590-641, Table of Contents, doi:10.1128/MMBR.00016-08.
12. Lee, D.G.; Jang, I.S.; Yang, K.E.; Yoon, S.J.; Baek, S.; Lee, J.Y.; Suzuki, T.; Chung, K.Y.; Woo, S.H.; Choi, J.S. Effect of rutin from tartary buckwheat sprout on serum glucose-lowering in animal model of type 2 diabetes. Acta pharmaceutica 2016, 66, 297-302, doi:10.1515/acph-2016-0021.”
Two more citations were added and all references were renumbered in revised manuscript.
3) Reviewer comment 3: The most important and necessary addition to Introduction part is to make logical their choice of investigated plants for this particular indication - why in TDD for diabetic therapy?
Response: The authors thank the eminent reviewer’s comment.
For diabetes treatment, there are 7 kinds of DM medicines, including sulfonylurea, meglitinide, biguanides, thiazolidinediones, dipeptidyl peptidase-4 inhibitor, α-glucosidase inhibitors, and sodium glucose co-transporters 2 inhibitor. However, numerous side effects such as gastrointestinal discomfort, edema, weight gain, and urinary tract infections are brought about clinically hypoglycemic drugs. Insulin injection is also used for treating diabetes particularly in TIDM and severe hyperglycemia. However, some patients cannot endure the injection due to severe hypoglycemia or dizzy even shock. In addition, insulin cannot be used by oral route because insulin loses its activity in stomach acids. So TDD is a preferred route of diabetic therapy and is convenient for patients who cannot take medicines easily or suffer from invasive cure. Therefore no-invasion treatment becomes more reliable and acceptable.
The plants used in the study are species with nettle stings at leaf blades. The plants contain some chemicals including histamine, oxalic acid, and tartaric acid playing different roles in inducing contact dermatitis.
We hypothesized that the crude extracts of three plants can disrupt the complex structure of SC allowing drug like insulin can be used as TDD to penetrate the damaged SC structure. The results confirmed our supposition based on the in vivo OGTT test. That was the reason why we concluded the extracts can play a role as the adjuvants in TDD.
In the submitted manuscript, we have described the reason why we choose plant extracts for TDD in diabetes and why crude extracts can be the potential adjuvants in the Introduction (page 2, line 64-68) and in the Discussion (page 9, line 275-284 of the submitted manuscript) as followings:
Introduction
“One of a vital problems in the insulin therapy is the paucity of patient consent. Therefore, TDD is a preferred route of diabetic therapy and convenient for patients who can’t take medicines easily [13] or suffer from invasive cure. However, the delivery reagents of TDD for insulin or insulin sensitizers (curcumin, rutin) in applicable to the problem of diabetic patients remain to be explored.”
Discussion
“For decades, the use of skin as a mean for drug transfer has been a fascinating preference to traditional approaches including intravenous and oral administration. However, human skin is arranged to be a superior permeability barrier to prevent entry of nearly all but small molecules [19]. So the exploration for the optimal skin penetration enhancer has been the spotlight of appreciable investigation exertion throughout the years. Herein, for the first time, we discovered that the natural sting crude extracts of Dm, Ut, and Ao impaired the structures of stratum corneum, stratum granulosum, and stratum spinosum of the mouse skin epidermis (Figure 2) to boost the skin permeation of insulin and insulin sensitizers (curcumin and rutin) for reducing the blood sugar level in diabetic ICR mice (Figure 4) while causing very slight skin erythema without skin edema (Figure 1) or any damage to HaCaT human skin keratinocytes (Figure 3).”
4) Reviewer comment 4: When first mentioning the full Latin name (species name, author name and Family, as well), should be provided for the plant species mentioned in the text.
Response: The authors thank the eminent reviewer’s comment.
The full Latin names of three plants have been added to the Introduction (page 2, line 69 and 72 of the revised manuscript) as follow:
“Dendrocnide meyeniana (Dm, Urticaceae) and Urtica thunbergiana Sieb. & Zucc (Ut, Urticaceae) are species with nettle stings at leaf blades formed by cells with high Si/Ca ratio.”
“Alocasia odora (Lodd.) Spach (Ao, Araceae) is a traditional herbal medicine for treating abdominal pain, cholera, hernia, stomach ache, wound healing, and fungi.”
5) Reviewer comment 5: In M&M section, it is necessary to describe the whole procedure of obtaining the extracts for investigation, giving the dry residue, as the dose of administration further in the experiments were given as concentration. In all performed experiments, authors are asked to provide the solvent in which the crude extracts of the investigated plants were dissolved. Besides, the preparation of curcumin and rutin solutions that were administrated in the experiments should be explained, as the solubility of those compounds is different.
Response: The authors thank the eminent reviewer’s comment.
The above information has now been added to the “Plant Materials and The Preparation of Sting Crude Extracts” section of the Materials and Methods (page 3, line 101-107 of the revised manuscript):
“Fresh leaves of Dm, Ut, and Ao were dried at room temperature (RT), cut into pieces, ground using a mill, sieved through a no. 40-mesh, and extracted by soaked with 50% ethanol/H2O at RT for one week. The extracts of Dm, Ut, and Ao were collected by filtration and vacuum evaporation. All extracts were stored at 4 °C avoiding light exposure. In all performed experiments, the crude extracts, curcumin, and rutin were dissolved in the distilled water and filtered by sterile syringe filter with a 0.45 µm pore size. Capsaicin was solubilized in 10% Tween 80, 10% ethanol, and 0.9% saline.”
6) Reviewer comment 6: In Experimental section, authors are asked to give the composition of negative control - “hand cream”, as stated in text (line 176).
Response: The authors thank the eminent reviewer’s comment.
The compositions of hand cream are including water, glycerine, cetearyl alcohol, stearic acid, sodium cetearyl sulfate, methylparaben, propylparaben, dilauryl thiodipropionate, and sodium sulfate. We purchased it from Neutrogena (Johnson & Johnson, USA).
The above information has now been added to the “Chemicals and Reagents” section of the Materials and Methods (page 2-3, line 92-94 of the revised manuscript):
“Hand cream (the compositions including water, glycerine, cetearyl alcohol, stearic acid, sodium cetearyl sulfate, methylparaben, propylparaben, dilauryl thiodipropionate, and sodium sulfate) was purchased from Neutrogena (Johnson & Johnson, USA).”
7) Reviewer comment 7: The main concern is the lack of positive control - as the enhancement effect of Ut, Dm and Ao extracts on in vivo insulin permeation through diet-induced diabetic mouse skin was investigated, the positive control should also be included in the experiment, meaning some of the synthetically obtained substances, for which clinically, or experimentally confirmed effects as enhancers of permeability in TDD exist in literature.
Response: The authors thank the eminent reviewer’s comment.
We fully agree with the Reviewer’s comment. My Ph.D. candidate, Yuh-Ming Fuh has been asked the same question as the Reviewer’s comment by the Committee. He has added Capsaicin as a positive control and tried doing in vivo experiments again.
We have added some information about Capsaicin in the revised manuscript:
Introduction (page 2, line 81-85):
“In this work, capsaicin isolated from Capsicum annuum (Solanaceae) was used as a positive control for penetration enhancer since it has shown analgesic advantages in postherpetic neuralgia, painful polyneuropathies, and surgical neuropathic syndromes [22]. Furthermore, the capsaicin 8% patch was approved by the Food and Drug Administration (FDA) for postherpetic neuralgia [23].”
References (page 12, line …):
“22. Jorge, L.L.; Feres, C.C.; Teles, V.E. Topical preparations for pain relief: efficacy and patient adherence. Journal of pain research 2010, 4, 11-24, doi:10.2147/JPR.S9492.
23. Groninger, H.; Schisler, R.E. Topical capsaicin for neuropathic pain #255. Journal of palliative medicine 2012, 15, 946-947, doi:10.1089/jpm.2012.9571.”
Two more citations were added and all references were renumbered in revised manuscript.
Materials and Methods (page 4, line 147):
“Ca (0.25 mg/mL)”
Figure 4A-D of the Results (page 8, line 258-259):
The legend of Figure 4 of the Results (page 9, line 264-266):
“Mouse skin was smeared with Dm (2 mg/mL), Ut (2 mg/mL), Ao (2 mg/mL), or Ca (0.25 mg/mL) with/without insulin (10 pM; 0.17 U), curcumin (20 μg/mL), or rutin (80 μg/mL), respectively.”
Section 3.4 of the Results (page 8, line 235, 238, and 244):
“or Ca at dose 0.25 mg/mL”, “or capsaicin compound”, “or capsaicin”.
Discussion (page 10, line 306):
“or with 0.25 mg/mL of Ca”.
8) Reviewer comment 8: Page 8, line 230, please, correct ug/mL into μg/mL
Response: The authors thank the reviewer for pointing this out.
We have made the necessary correction (Results; page 8, line 250 in the revised manuscript): “ug/mL” à “μg/mL”.
9) Reviewer comment 9: Page 9, Figure 4, please, give the explanation for Rut op, Cur op in the legend of Figure 4.
Response: The authors thank the reviewer for pointing this out.
We have corrected the figure 4E-H and given the explanation for Rut po, Cur po (Rut per os, Cur per os) in the legend of Figure 4 of the Results, page 9, line 270-271 in the revised manuscript):
“Cur po: oral administration of curcumin”
“Rut po: oral administration of rutin”
Again, we greatly appreciate all of your helpful and constructive comments. We hope the eminent reviewer can accept our extensive response. We look forward to your reply in your convenient time.
Sincerely yours,
Ching-Feng Weng, Ph.D., Professor
Department of Life Science and Institute of Biotechnology
National Dong Hwa University
Shoufeng, Hualien 97401, Taiwan
Phone: +886-3-890-3637

Round 2
Reviewer 2 Report
Manuscript Number: medicina-468412
Title: Sting Plant Extracts as Penetration Enhancers Potentiate Transdermal Delivery of Hypoglycemic Compounds
Comments to Authors
Recommendation: Publish with minor changes
The authors of the article "Sting Plant Extracts as Penetration Enhancers Potentiate Transdermal Delivery of Hypoglycemic Compounds” mainly responded to the suggested comments.
However, still remains the question regarding the solubility of curcumin in water. Curcumin has been known not only for insolubility in water, but for its hydrophobicity, as well. Even, if authors rise temperature in order to increase the solubility, or change pH, and afterwards filtered it, the concentration of the curcumin in the obtain solution could not be as the stated in the text. Please, provide the sufficient explanation how the used concentration in the experiments for curcumin was determined.
Perhaps, the useful text might be:
November 2012, The Journal of Physical Chemistry B 116(50), DOI: 10.1021/jp3050516
Temperature-Dependent Spectroscopic Evidences of Curcumin in Aqueous Medium: A Mechanistic Study of Its Solubility and Stability
Author Response
Manuscript ID: Medicina-468412
We greatly appreciate the helpful comments of the Reviewer 2. We have made the necessary changes, which we believe substantially improved the manuscript. In response to the Reviewer 2’s comments, we have added new sentences and edited existing sentences in the revised version according to the comments. All of the changes are yellow highlighted in the revised manuscript. Our point-by-point responses are set out below.
Responses to specific comments from Reviewer #2
The general comment of Reviewer #2:
The authors of the article "Sting Plant Extracts as Penetration Enhancers Potentiate Transdermal Delivery of Hypoglycemic Compounds” mainly responded to the suggested comments.
Specific comments:
Reviewer comment 1: However, still remains the question regarding the solubility of curcumin in water. Curcumin has been known not only for insolubility in water, but for its hydrophobicity, as well. Even, if authors rise temperature in order to increase the solubility, or change pH, and afterwards filtered it, the concentration of the curcumin in the obtain solution could not be as the stated in the text. Please, provide the sufficient explanation how the used concentration in the experiments for curcumin was determined.
Perhaps, the useful text might be:
November 2012, The Journal of Physical Chemistry B 116(50), DOI: 10.1021/jp3050516
Temperature-Dependent Spectroscopic Evidences of Curcumin in Aqueous Medium: A Mechanistic Study of Its Solubility and Stability
Response: The authors thank the eminent reviewer’s comment.
We are really appreciated for the recommended paper from the Reviewer.
We have tried to dissolve curcumin in distilled water with the same protocol as your suggested paper on The Journal of Physical Chemistry B 116(50), DOI: 10.1021/jp3050516 before. After heating to 70oC, we found that there were some precipitates in the solution. So we decided to use ethanol instead of distilled water to dissolve curcumin.
We are terribly sorry for the mistake in the revised manuscript.
We have added the above information in the Plant Materials and The Preparation of Sting Crude Extracts section of the Materials and Methods (page 3, line 105-108 of the revised manuscript):
“In all performed experiments, the crude extracts and rutin were dissolved in the distilled water while curcumin was dissolved in ethanol. Capsaicin was solubilized in 10% Tween 80, 10% ethanol, and 0.9% saline. The prepared samples were filtered by sterile syringe filter with a 0.45 µm pore size.”
We greatly appreciate your helpful comment and kind suggestion. We hope the eminent reviewer can accept our extensive response. We look forward to your reply in your convenient time.
Sincerely yours,
Ching-Feng Weng, Ph.D., Professor
Department of Life Science and Institute of Biotechnology
National Dong Hwa University
Shoufeng, Hualien 97401, Taiwan
Phone: +886-3-890-3637
